# Transcriptome and Small-RNA Sequencing Reveals the Response Mechanism of *Brassica napus* to Waterlogging Stress

**DOI:** 10.3390/plants14091340

**Published:** 2025-04-29

**Authors:** Xianshuai Song, Lan Ge, Kaifeng Wang, Nian Wang, Xinfa Wang

**Affiliations:** 1Key Laboratory of Biology and Genetic Improvement of Oil Crops, Ministry of Agriculture and Rural Affairs, Oil Crops Research Institute of the Chinese Academy of Agricultural Sciences, Wuhan 430062, Chinamapledrive@163.com (K.W.); 2Hubei Hongshan Laboratory, Wuhan 430070, China

**Keywords:** *Brassica napus*, waterlogging, comparative transcriptome, microRNA

## Abstract

Rapeseed (*Brassica napus*) is highly susceptible to waterlogging during the seedling stage; however, most of the studies on its gene expression under waterlogging stress have focused on transcriptional regulation, with little work conducted on post-transcriptional regulation to date. To elucidate this regulatory network, comparative transcriptome and miRNA analyses in the leaves and roots of rapeseed Zhongshuang11 (ZS11) were performed. Differentially expressed genes (DEGs) and miRNAs (DEmiRNAs) were identified by comparing the normal planting condition (the control group, CKT) with waterlogging treatment (WLT). DEGs identified in leaves and roots were enriched in different metabolic pathways, indicating their distinct mechanisms in response to waterlogging stress. In total, 68 and 82 DEmiRNAs were identified in leaves and roots, respectively, predicted to target 543 and 2122 DEGs in each tissue. Among these, 12 and 9 transcription factors (TFs) were exclusively targeted by DEmiRNAs in leaves and roots, respectively. Notably, six upregulated TFs in leaves were associated with the ethylene response and were predicted targets of bna-miR172 family members, and four TFs in roots participated in the ethylene response pathway. Furthermore, bna-miR169, along with novel-miR-23108 and novel-miR-42624 family members, played crucial roles in waterlogging response of rapeseed. Combining with the determination results of ethylene and jasmonic acid content, a preliminary model of miRNA-mediated gene expression regulation in rapeseed response to waterlogging stress was developed. These findings advance our understanding of transcriptional regulation under waterlogging and lay a theoretical foundation for improving rapeseed waterlogging tolerance.

## 1. Introduction

Global climate change has intensified extreme weather events, particularly heavy rainfall and flooding, causing widespread waterlogging in agricultural fields. As one of the major abiotic stresses, waterlogging severely threatens crop productivity and global food security [1]. Rapeseed (*Brassica napus*) is one of the most important economic crops and oil crops [2]; however, rapeseed is highly sensitive to waterlogging stress. In China, rapeseed is predominantly cultivated in the Yangtze River region, where a rice–rapeseed rotation planting system has always been conducted. The preceding cultivation of rice and frequent rainfall elevates the ground water levels, which leads to higher requirements for improving the waterlogging tolerance of rapeseed [3,4].

The rapeseed shows exceptional sensitivity to waterlogging stress during the seedling stage, when just 3 days of submergence significantly compromises yield potential, escalating to 20–50% losses under prolonged (14 day) stress conditions [5], underscoring the critical need to understand seedling stage tolerance mechanisms. However, current research on waterlogging response mechanisms in rapeseed during the seedling stage is limited and has primarily focused on transcriptional regulation [6]. Transcriptomic analyses of waterlogged rapeseed seedlings have revealed that genes respond to waterlogging stress mainly via enrichment in key pathways, including phenylpropanoid biosynthesis, hormone signal transduction, starch and sucrose metabolism, and reactive oxygen species (ROS) detoxification [7,8,9]. Moreover, the aerial parts of waterlogged rapeseed seedlings show rapid downregulation of photosynthesis-related genes and upregulation of genes associated with senescence and stress responses [10].

Ethylene (ET) and jasmonic acid (JA) are essential plant growth regulators known to mediate defense responses under abiotic stress [11,12]. ET promotes the formation of adventitious roots in soybean [13] and cucumber [14] under waterlogging; thus, it alleviates hypoxia stress and maintains normal root metabolism and function. In contrast, JA inhibits waterlogging-induced adventitious root (AR) formation in cucumber [15]. Similarly, studies have shown that spraying methyl jasmonate on leaves can increase ethylene levels, helping to alleviate waterlogging stress [16].

Emerging evidence highlights the critical role of post-transcriptional regulation—particularly by small RNAs such as microRNAs (miRNAs)—in plant stress adaptation [17,18,19,20,21], such as drought stress [22], heat stress [23], and salt stress [24]. miRNAs, a class of endogenous non-coding small RNAs ranging from 18 to 36 nucleotides, have emerged as crucial regulators of plant development and stress responses. They function by guiding target mRNA cleavage, translational inhibition, or influencing epigenetic modifications [25]. For example, miR172 and its target genes act as key regulators not only in the phase transition of plants but also in flower organ development and responses to environmental stresses [26,27]. Meanwhile, miR172 can regulate AP2-like transcription factors (TFs) to participate in plant responses to various environmental stresses [28]. The miR169 family is an evolutionarily conserved miRNA family in plants, and the miR169/NF-YA module plays a key role in plant development and responses to abiotic stress [29]. However, the roles of small RNAs in gene expression regulation in rapeseed under waterlogging have been seldom studied.

Roots, as the primary organs for water and nutrient acquisition in plants, play a crucial role in plant growth and development [30]. Waterlogging inhibits root respiration and significantly reduces the energy status of root cells [31], and leads to reduced root vitality and impairs the uptake and transport of water and nutrients to the aerial parts of the plant. In rapeseed, hypoxia–reaeration cycles severely reduce root hydraulic conductivity, likely due to oxidative stress and aquaporin (*BnPIP2s*) downregulation [32]. The more waterlogging-tolerant rapeseed cultivar maintains greater root cell wall thickness and pectin content, linked to upregulated *PGIP2* genes that may reduce pectin degradation [33]. Compared with the direct suppression impact of waterlogging on roots, the impact of waterlogging on leaves is indirect. Waterlogging stress will reduce the activity of photosynthesis-related enzymes, hinder the chlorophyll synthesis and promote its degradation, and cause leaf wilting, chlorosis, senescence, and abscission, ultimately inhibiting plant growth and development and further leading to plant death [15]. The response of rapeseed roots and leaves to waterlogging stress should be somewhat different; however, comparative analyses of their gene regulatory mechanisms remain limited.

In order to identify key genes and reveal the regulatory pathways involved in waterlogging at the seedling stage of rapeseed, the rapeseed accession ZS11 was used as a model plant. Physiological and biochemical indicators under waterlogging were measured; ET and JA were found to play a key role in response to waterlogging stress in rapeseed; transcriptome and small-RNA sequencing were conducted to identify DEGs and DEmiRNA in leaves and roots, respectively; the regulation relationship between DEmiRNA and DEGs was predicted; and the gene expression regulatory network was preliminarily constructed. This study will deepen our understanding of the gene regulation mechanisms of rapeseed in response to waterlogging stress.

## 2. Results

### 2.1. Waterlogging Inhibits Growth and Alters Physiology of Rapeseed

ZS11, a widely planted rapeseed inbred line in China known for its excellent comprehensive traits, served as a model accession in this study. The ZS11 plants were subjected to two conditions: normal planting (the control group, CKT) and waterlogging treatment (the waterlogging group, WLT). After 4 days of waterlogging, WLT plants developed chlorosis in new leaves while the older leaves exhibit yellowing or purpling; after 8 days of waterlogging, older leaves displayed advanced senescence and abscission, with new leaves exhibiting severe chlorosis (Figure 1A). With the prolongation of waterlogging time, the net photosynthetic rate, the stomatal conductance, the SPAD values, and the chlorophyll content of ZS11 leaves of WLT gradually declined compared to the CKT, with significant differences observed on the 4th day (Figure 1B,C). Waterlogging also inhibited the growth of fresh and dry weights in both leaves and roots, with significant differences between CKT and WLT emerging between the 4th and 6th days (Figure 1D,E). Additionally, soluble sugar and malondialdehyde (MDA) content in leaves increased under waterlogging (Figure 1F,G). Compared to the CKT, ethylene precursor ACC levels increased significantly after 8 days of waterlogging, whereas jasmonic acid levels exhibited a decreasing trend (Figure 1H,I). These findings demonstrate that rapeseed growth was seriously inhibited when subjected to waterlogging for more than 4 days.

### 2.2. DEGs Identified in Leaves and Roots Under Waterlogging Stress

Based on data analysis and observations, the inhibition of waterlogging on rapeseed growth became highly pronounced by the 8th day. Consequently, leaves and roots of ZS11 from both the WLT and CKT on 8th days were selected for comparative transcriptome analysis. Principal component analysis (PCA) based on the gene expression data revealed that the three replications of leaves or roots of WLT or CKT were clustered together, respectively, demonstrating the good reproducibility of the RNA-seq data (Appendix A).

A total of 12,185 DEGs were identified in leaves, including 5662 downregulated and 6523 upregulated genes; while 19,840 DEGs were identified in roots, including 10,927 downregulated and 8913 upregulated genes (Appendix A, Appendix A). Only 4058 DEGs were shared between leaves and roots, with root-specific DEGs (15,782) being approximately twice as numerous as leaf-specific DEGs (8127) (Appendix A, Appendix A). Gene Ontology (GO) enrichment analysis revealed that DEGs common to both leaves and roots were primarily associated with the regulation of jasmonic-acid-mediated signaling pathways, the ethylene-activated signaling pathway, golgi vesicles, the extracellular matrix, calcium ion binding, and hormone binding (Figure 2A). Leaf-specific DEGs were enriched in processes such as glycosylation, glycoprotein metabolic process, response to endoplasmic reticulum stress, golgi medial cisterna, integral component of endoplasmic reticulum membrane, and beta-1,4-mannosyltransferase activity (Figure 2B). In leaves, the glycoprotein metabolic pathway was significantly enriched (*p* < 0.01, Appendix A); it is known to produce antimicrobial compounds such as UDP-glucose:glycoprotein glucosyltransferase. Notably, the genes involved in regulating the synthesis of UDP-glucose:glycoprotein glucosyltransferase were all downregulated (Appendix A), suggesting a reduced capacity to resist waterlogging stress. Root-specific DEGs were linked to the phenylpropanoid metabolic process, regulation of the jasmonic-acid-mediated signaling pathway, cellular response to ethylene stimulus, extrinsic component of membrane, and inorganic anion transmembrane transporter activity (Figure 2C, Appendix A). The enrichment of the phenylpropanoid biosynthesis pathway (*p* < 0.01, Appendix A) suggests activated defense responses, as this pathway produces antimicrobial compounds like lignin and flavonoids. Notably, PAL (phenylalanine ammonia-lyase) genes showed a 3–5-fold upregulation (Appendix A).

KEGG pathway analysis showed that DEGs common to both roots and leaves were enriched in pathways related to glycerolipid metabolism, amino sugar and nucleotide sugar metabolism, sulfur metabolism, and fatty acid degradation (Figure 3A). DEGs regulating glycerol-3-phosphate acyltransferase were significantly upregulated. As glycerolipid metabolism critically regulates cellular unsaturated fatty acid levels, this suggests that enhanced phospholipid biosynthesis may maintain membrane homeostasis during stress. Leaf-specific DEGs were primarily associated with *N*-Glycan biosynthesis, lipid biosynthesis proteins, and DNA replication (Figure 3B). Root-specific DEGs were predominantly enriched in phenylpropanoid biosynthesis, ubiquinone and other terpenoid-quinone biosynthesis, and photosynthesis-antenna proteins (Figure 3C, Appendix A).

### 2.3. Small RNAs Identified in Leaves and Roots Under Waterlogging Stress

To explore miRNA regulatory mechanisms in rapeseed under waterlogging stress, small RNA sequencing was also conducted, and small RNAs were identified by aligning the clean reads to the Rfam and Repbase databases using Bowtie software. These small RNAs consisted of ribosomal RNA (rRNAs; 55.24% and 51.66% in leaves and roots, respectively), transfer RNA (tRNAs; 12.94% and 15.09%), small nuclear RNA (snRNAs; 0.54% and 0.29%), repeat sequences (21.96% and 31.19%), and unknown fragments (32.33% and 32.8%) (Table 1). Unlike housekeeping ncRNAs (rRNAs, tRNAs, and snRNAs) that maintain relatively stable expression levels, miRNAs serve as regulatory ncRNAs that modulate transcription and translation. These stress-responsive molecules, which play crucial roles in plant stress adaptation, were the focus of our investigation.

In roots, the majority of miRNAs (22.35%) exhibited a sequence length of 24 nucleotides (nt), followed by 19 nt (21.11%) and 30 nt (19.72%) (Figure 4A). Similarly, in leaves, the majority of miRNAs had a sequence length of 24 nt (26.95%), followed by 26 nt (26.75%) and 21 nt (21.58%) (Figure 4B, Appendix A). The miRNA length distribution ranging from 18 to 30 nt was consistent with the typical base length distribution of miRNAs in plants. Dicer enzyme exhibits a strong “U” bias at the first nucleotide of the 5′ end when recognizing and cleaving precursor miRNAs. This base preference is used to further validate the identified known and novel miRNAs. The known miRNAs had a sequence length ranging from 20 to 24 nt, with most miRNAs having a sequence length of 21 nt in both the roots (72.4%) and leaves (72.06%) (Appendix A). The novel identified miRNAs had a sequence length ranging from 18 to 25 nt, and most novel miRNAs had a sequence length of 24 nt (50.7% and 36.49%), followed by 21 nt (27.9% and 31.08%) in the leaves and the roots, respectively (Appendix A, Appendix A). The first base of the 24 nt novel miRNAs was predominantly the “A” in leaves and roots (Appendix A).

### 2.4. Annotation of miRNA Families

miRNAs annotated in the public miRBase database with typical precursor secondary structures are defined as known miRNAs, while those lacking miRBase annotations but meeting Dicer cleavage site characteristics are defined as novel identified miRNAs. miRNAs are highly conserved among different species, and miRNAs were classified into miRNA families based on sequence similarity. Finally, a total of 127 miRNAs were identified in leaves under CKT and WLT, classified into 45 miRNA families. This included 81 known miRNAs (assigned to 23 families) and 46 novel miRNAs (assigned to 27 families) (Appendix A). Notably, both the known and novel miRNAs were assigned to five common families: MIR156, MIR164, MIR393, MIR394, and MIR399 families (Appendix A). In roots, 108 miRNAs were identified and classified into 35 families, including 82 known miRNAs (assigned to 24 families) and 26 novel miRNAs (assigned to 13 families) (Appendix A). Both the known and novel miRNAs were assigned to two common families: MIR156 and MIR399 families (Appendix A).

A total of 27 miRNA families were identified both in leaves and roots, and 21 and 7 miRNA families were only identified in leaves or roots, respectively (Appendix A). MIR169_1, with 10 family members, had the highest number of family members among the known miRNA families both in leaves and roots. MIR5654 and mir-86, each with five family members, had the highest number of members among the novel miRNA families in leaves (Figure 4C,D); while MIR158, MIR156, and MIR399, each with four members, had the most family members among the novel miRNA families in roots (Figure 4E,F). Among the leaf-specific miRNA families, MIR164 was known to be involved in leaf morphogenesis, lateral root development, and organ abscission [34]; and MIR394, MIR393, MIR477, and MIR2118 were known to participate in the regulation of plant growth, development, and responses to environmental stresses [35,36,37,38]. Among the root-specific miRNA families, mir-25 and mir-144 were known to be involved in regulating cell proliferation, apoptosis, and differentiation [39,40].

### 2.5. Differentially Expressed miRNAs Under Waterlogging Stress in Rapeseed

To identify waterlogging-responsive miRNAs in rapeseed, differentially expressed miRNAs (DEmiRNAs) between CKT and WLT were analyzed. A total of 68 DEmiRNAs, 51 upregulated and 17 downregulated, were identified in leaves (Appendix A), and 2 upregulated DEmiRNAs (bna-miR2111c and novel_miR_17706) had a fold change greater than 8, while 2 downregulated DEmiRNAs (bna-miR172b and bna-miR172c) had a fold change greater than 3 (Appendix A). A total of 82 DEmiRNAs, 48 upregulated and 34 downregulated, were identified in roots (Appendix A), and 7 upregulated DEmiRNAs (bna-miR2111a-3p, bna-miR2111b-3p, bna-miR2111c, novel_miR_4376, novel_miR_27016, novel_miR_30618, and novel_miR_19054) had a fold change greater than 7, and the 2 downregulated DEmiRNAs (novel_miR_34897 and novel_miR_19605) had a fold change greater than 4 (Appendix A).

A total of 69 and 55 unique DEmiRNAs were identified in roots and leaves, respectively, and only 13 DEmiRNAs were identified in both roots and leaves, in which 8 DEmiRNAs (bna-miR399a, bna-miR399b, bna-miR2111a-3p, bna-miR2111a-5p, bna-miR2111b-3p, bna-miR2111b-5p, bna-miR2111c, and bna-miR2111d) were upregulated, and 5 DEmiRNAs (bna-miR172b, bna-miR172c, bna-miR395d, bna-miR395e, and bna-miR395f) were downregulated. miR172, miR395, miR2111, and miR399 are conserved miRNA families in plants, playing key roles in plant growth, development, and stress responses [41,42,43,44]. These DEmiRNAs may play an essential role in the response to waterlogging stress of rapeseed, but their functions require further investigation.

### 2.6. GO and KEGG Analysis of DEmiRNA Target Genes

Through miRNA-mRNA target gene predictions, a total of 543 DEGs were predicted to be targeted by 68 DEmiRNA in leaves, and 2122 DEGs were predicted to be targeted by 82 DEmiRNA in roots (Appendix A). GO analysis showed that DEGs predicted to be targeted by the 68 DEmiRNA in leaves were primarily related to positive regulation of cell cycle phase transitions, hydrogen sulfide biosynthesis processes, ribonucleotide nuclease complexes, external components, and NADPH dehydrogenase activity (Figure 5A). DEGs predicted to be targeted by the 82 DEmiRNA in roots were primarily related to sucrose metabolism, nucleic acid transport, cytoplasmic side of the plasma membrane, endogenous components of the mitochondrial inner membrane, and triphosphate (ATP) activity (Figure 5B, Appendix A).

Additionally, 62.1% (337/543) of the DEGs were targeted by DEmiRNAs novel_miR_23108 in leaves; and 11.0% (234/2122) of the DEGs were targeted by DEmiRNAs novel_miR_42624 in roots. GO enrichment analysis revealed that the DEGs targeted by novel_miR_23108 were primarily enriched in the negative regulation of response to biotic stimulus, negative regulation of response to external stimulus, intrinsic component of mitochondrial inner membrane, and signaling receptor binding (Figure 5C). DEGs targeted by novel_miR_42624 were primarily enriched in the glutamine family amino acid metabolic process, ethylene-activated signaling pathway, cellulose biosynthetic process, and carboxylic acid transmembrane transporter activity (Figure 5D, Appendix A). Additionally, among the 37 DEGs regulated by differentially expressed bna-miR172b identified in leaves, 16 were identified as AP2-type TFs. In roots, differentially expressed bna-miR172a was predicted to target 33 DEGs, while bna-miR172b was predicted to targeted 23 genes (Appendix A).

### 2.7. Transcription Factors in the DEmiRNAs

TFs play a key role in gene expression regulation. In this study, 12 and 9 TFs were contained among the DEGs targeted by DEmiRNA in leaves and roots, respectively (Appendix A). Notably, no TFs were found to be common to both leaves and roots. Among the 12 TFs identified in leaves, 6 TFs (3 RAP2-7 TFs, 2 TOE2 TFs, and 1 SMZ TFs) were involved in the ethylene-responsive pathway, all of which were upregulated under waterlogging and were predicted to be targeted by bna-miR172 family members (Appendix A). In roots, the nine TFs (two MYC TFs, two ERF TFs, one AT-GTL1 TF, one NAC053 TF, one EIN3 TF, one BZIP28 TF, one ABF2 TF) were mainly involved in ethylene and jasmonic acid metabolism and synthesis (Appendix A). The RT-qPCR results of these 21 TFs showed a consistent results with the transcriptomic data, indicating the reliability of the RNA-seq results (Figure 6A,B). These TFs may play an important role in inducing the expression of ET and JA under waterlogging stress.

## 3. Discussion

In this study, when rapeseed was subjected to waterlogging for more than 4 days, its growth was significantly inhibited (Figure 1). After 8 days of waterlogging, the changes of some indicators such as chlorophyll content, SPAD values, and soluble sugar content began to stabilize, suggesting that the rapeseed plants had likely reached their peak response to waterlogging stress at this stage. Therefore, samples collected on day 8 were used for comparative transcriptomic and miRNA analyses.

Waterlogging stress triggers oxidative responses, with soluble sugars and MDA serving as key physiological indicators. Soluble sugars serve dual roles as osmoprotectants maintaining osmotic homeostasis under hypoxia and as metabolic substrates supporting anaerobic respiration. Conversely, MDA—a lipid peroxidation byproduct—quantitatively reflects oxidative membrane damage, where elevated levels indicate greater waterlogging-induced stress [45,46]. In this study, the MDA content and soluble sugar content increased significantly with the extension of waterlogging duration (Figure 1F), which is consistent with previous findings [47].

Consistent with the findings of Ksouri et al. (2016) [48], a greater number of DEGs in waterlogged roots compared to leaves in rapeseed were identified. We hypothesized that this pattern may result from either the inherently higher sensitivity of roots to waterlogging stress, or the direct exposure of roots to waterlogging versus the indirect systemic effects on leaves. Profiling of DEGs under waterlogging stress, followed by functional and pathway enrichment analyses, provided a comprehensive overview of the regulatory responses involved. In this study, the enriched pathways of DEGs differed between leaves and roots. Leaf-specific DEGs were primarily enriched in processes such as glycosylation, cellular redox homeostasis maintenance, and response to endoplasmic reticulum stress, while root-specific DEGs were mainly enriched in phenylpropanoid metabolic process, regulation of the jasmonic-acid-mediated signaling pathway, and cellular response to ethylene stimulus. These waterlogging-responsive pathways enriched in rapeseed were also identified in other crops, demonstrating evolutionary conservation [49,50,51].

Currently, comparative transcriptomes or miRNA analysis have been widely applied in studying waterlogging tolerance in plants such as rapeseed [52], wheat [53], maize [54,55], and cucumber [56]. The combination of transcriptome and miRNA analysis can provide us with a more comprehensive understanding of gene expression regulation to abiotic stress [57], and it enables the identification of novel regulatory factors [58]. In this study, a total of 68 and 82 DEmiRNA in leaves and roots were identified, respectively, and 543 DEGs in leaves and 2122 DEGs in roots were predicted to be targeted by these DEmiRNAs. Only 13 DEmiRNAs were identified both in leaves and roots, and DEGs identified in leaves and roots were enriched in different metabolic pathways, also indicating distinct mechanisms in response to waterlogging stress. Twelve and nine TFs were found in DEGs targeted by DEmiRNAs in leaves and roots, respectively. Among the 12 TFs identified in leaves, the 6 upregulated TFs were predicted to be targeted by bna-miR172 and were involved in ethylene biosynthesis; among the 9 TFs identified in roots, 4 were involved in ethylene metabolism-related pathways. Our results demonstrated that most DEmiRNAs were upregulated, a pattern consistent with previous findings in cucumber and maize [59,60], suggesting that the upregulation of miRNAs may play an important role in rapeseed’s response to waterlogging stress.

In leaves, 62.1% (337/543) of the DEGs were predicted to be targeted by the DEmiRNA novel-miR-23108, and these DEGs were enriched in biotic and external stimulus responses (Appendix A). In roots, 11.0% (234/2122) of the DEGs were predicted to be targeted by the DEmiRNA novel-miR-42624, and these DEGs were enriched in ethylene-activated signaling (Appendix A). These two newly identified miRNAs may play an essential role in the waterlogging response of rapeseed in leaves and roots, respectively. We utilized the stem-loop RT-qPCR technique to measure the expression levels of novel-miR-23108 and novel-miR-42624, and the results were consistent with those obtained from small RNA sequencing. Additionally, MIR169-1 was the most abundant miRNA family identified in both leaves and roots of ZS11 (Figure 4C,E), and two MIR169-1 family members (bna-miR169m and bna-miR169a) were differentially expressed (Appendix A).

ET and JA are basic plant growth regulators that are known to be involved in the defense response produced by abiotic stress [11,12]. Flooding has been found to directly or indirectly activate the expression of *ACO5* and *ACS* genes in *Arabidopsis,* leading to increased ET biosynthesis [61]. Under waterlogging stress, jasmonic acid (JA) content exhibits tissue-specific variation. For example, under flooding stress, JA levels in citrus leaves increased significantly compared to the control, while JA concentrations in the roots decreased sharply. The reduction in root JA content may be attributed to antagonistic crosstalk between ET and JA, which maintains hormone balance in plant responses to waterlogging, ensuring cellular homeostasis [62]. In this study, a combination of enrichment analysis of DEGs, analysis of DEmiRNAs targeted genes, and functional assessment of key TFs all revealed connections between ET and JA metabolic pathways and response to waterlogging stress of rapeseed.

Based on the above analysis, a proposed gene expression regulatory network for rapeseed under waterlogging stress was constructed (Figure 7). Key miRNAs, including novel-miR-42624, novel-miR-23108, bna-miR172, and bna-miR169, were identified to target TFs such as *RAP2-7*, *SMZ*, *TOE2*, *ERF*, *WRKY*, *MYC*, *NAC*, *MYB,* and *TAF12B*, thereby modulating the synthesis and metabolism of ET and JA. These regulatory networks integrate multiple stress-response pathways, including NADPH dehydrogenase activity, ethylene-activated signaling, phosphotransferase activity, and jasmonic-acid-mediated signaling pathway. The critical miRNA-gene and miRNA-TF interactions identified in this study lay the groundwork for further investigation into the molecular mechanisms of rapeseed response to waterlogging stress. These key miRNAs and DEGs show promising potential for improving rapeseed waterlogging tolerance through molecular breeding. In combination with QTL mapping results, they can be used to develop more reliable MAS markers. Ultimately, gene editing technologies can be employed to validate the function of these miRNAs and DEGs, facilitating the improvement of waterlogging tolerance of rapeseed germplasm.

## 4. Materials and Methods

### 4.1. Materials and Waterlogging Treatment

The experiment was conducted indoors in a glasshouse condition (16 h light:8 h dark, 25 °C, 5000 lux). The test material was the *B. napus* cultivar ZS11. Healthy, full-grain, and uniform rapeseed seeds were selected and sown into 8 L nutrient pots, with a total of 200 seeds sown. Hoagland nutrient solution was used as the nutrient source for seed germination and seedling growth. Seeds were cultured in the dark for two days until the radicles emerged, after which the covers were removed, and the seedlings were grown under a light–dark cycle of 16:8 h, a temperature of 25 °C, and a relative humidity of 80%. After 5 days of cultivation, the seedlings were transplanted into 5 × 10 soil trays containing nutrient soil purchased from the laboratory. Once the seedlings had grown to the two to three leaf stage, waterlogging treatments were initiated. For waterlogging treatments, water level was maintained about 1 cm above the soil surface. Observations and sampling were conducted at 0, 2, 4, 6, 8, and 10 days of both CKT and WLT.

### 4.2. Measurement of Physiological and Biochemical Indices

For physiological measurements, the second fully expanded leaf from the top was selected at 0, 2, 4, 6, 8, and 10 days. The net photosynthetic rate and stomatal conductance were measured using an LI-6800XT photosynthesis system (LI-COR Biotechnology Co., Ltd., Beijing, China) [63]. SPAD values were determined using an SPAD-502 Plus device (Konica Minolta, Shanghai, China), and chlorophyll was extracted from plant tissue using 80% acetone [64]. Soluble sugar and MDA contents were determined using reagent kits from Solarbio Biotechnology Co., Ltd., (Beijing, China) with catalog numbers BC0030 (soluble sugar) and BC0025 (MDA). The ethylene precursor ACC and jasmonic acid (JA) contents in leaves and roots were measured using ELISA assay kits (ACC: NM-EL36135P, JA: NM-EL0887P) from NorminKoda Biotechnology Co., Ltd. (Wuhan, China). ET content was assessed based on ACC levels. Each treatment included three biological replicates. For biomass measurement, three plants with uniform growth were selected to determine the fresh weight of the shoots and roots (washed clean with water). The oven was preheated to 105 °C, and the weighed fresh plant material was placed into the oven and heated at 105 °C for 2 h. Then, the oven temperature was reduced to 80 °C and the plant material was dried to a constant weight before the dry mass was weighed [65].

### 4.3. Transcriptome Data Analysis

Leaves and roots were sampled from plants under waterlogging for 8 days. Transcriptome sequencing was performed on an Illumina HiSeq X platform (Zhenyue Biotechnology Co., Ltd., Wuhan, China). After implementing quality control measures, high-quality reads were mapped to the ZS11 reference genome (https://yanglab.hzau.edu.cn/BnIR/germplasm_info?id=ZS11.v0, accessed on 24 November 2024) using HISAT2 (v2.1.0) [66]. Read counts for each gene were calculated using HTSeq (v0.6.1) [67]. Differential expression analysis was performed based on expression levels across different sample groups. Genes with a |log_2_(fold change, FC)| ≥ 1 and *p*-value < 0.05 were identified as DEGs. Transcript abundance was quantified using TPM (transcripts per million).

### 4.4. Small RNA Data Analysis

After sequencing, raw reads were filtered to remove 3′ adapter sequences, trim low-quality bases at the 3′ ends (quality score < 20), exclude reads with an unknown base (N) proportion exceeding 10%, discard reads that were too short (<18 nt) or too long (>30 nt), and remove reads containing poly A/T/G/C stretches. The Q20, Q30, and GC content of the reads were calculated for subsequent analysis. Clean reads were aligned using Bowtie2 (v2.5.2) software [68] to the Rfam (https://rfam.org/, accessed on 12 January 2025) and Repbase (https://www.girinst.org/repbase/, accessed on 12 January 2025) databases to filter non-coding RNAs (ncRNAs), including ribosomal RNA (rRNA), transfer RNA (tRNA), small nuclear RNA (snRNA), small nucleolar RNA (snoRNA), and repetitive sequences. Unannotated reads were retained for further miRNA analysis. These unannotated reads and non-repetitive sequences were aligned to the ZS11 reference genome using Bowtie2 (v2.5.2) software, and their alignment information was assessed for quality.

### 4.5. miRNA Classification and Target Gene Prediction

Filtered reads were classified to analyze the composition of the small RNA dataset. Reads of 18–30 nt in length were selected as valid data for further analysis. Reads aligned to the reference genome were further compared with the mature sequences of known miRNAs in the miRBase database (http://www.mirbase.org/), including 2 nt upstream and 5 nt downstream of the miRNA sequence, allowing for a maximum of one mismatch. Potential precursor sequences were identified using the genomic alignment information of reads with the miRDeep2 (v2.0.0.8) software package [69]. Precursor sequences were scored using a Bayesian model based on the characteristics of miRNA generation (mature, star, and loop distribution), precursor structural energy (RNA fold and randfold), and read distribution, enabling the prediction of novel miRNAs.

Target gene prediction for both known and novel predicted miRNAs was performed using psRobot (V1.2) [70] based on the gene sequence information of ZS11. Read counts for miRNAs in each sample were quantified, followed by statistical analysis of expression data to identify genes with significantly different expression levels under various conditions. The DEmiRNAs analysis involved three main steps: Normalizing raw read counts, primarily to account for sequencing depth; calculating the probability (*p*-value) using a statistical model for hypothesis testing; and adjusting for multiple hypothesis testing to determine the false discovery rate (FDR), with padj as a common form. The R package DESeq (V2.10) [71] was used for statistical analysis, and the *p*-value was adjusted based on padj, with an adjusted *p*-value threshold of 0.05 and |log_2_(fold change, FC)| ≥ 1 to identify DEmiRNAs.

### 4.6. GO and KEGG Analysis

The DEGs identified from the transcriptome and miRNA analyses were subjected to Gene Ontology (GO) functional analysis, including GO functional enrichment and GO functional clustering (https://yanglab.hzau.edu.cn/BnIR/GO, accessed on 15 March 2025). Additionally, KEGG analysis was performed using the BnIR website (https://yanglab.hzau.edu.cn/BnIR/KEGG, accessed on 20 March 2025).

### 4.7. Real-Time Quantitative PCR Analysis

Real-time quantitative PCR (RT-qPCR) was used to confirm DEGs. Stem Loop RT-PCR was user to confirm DEmiRNAs. Primers of genes were designed using SnapGene (V6.2.1) software; primers of DEmiRNAs were designed using miRNA Design V1.01; *Actin7* and U6 were used as the internal control [72] (Appendix A). Relative expression levels were calculated using the method described by Livak and Schmittgen [73]. Total RNA was extracted from samples using the Plant RNA Extraction Kit (Omega, Norcross, CA, USA) according to the instructions. The integrity of the total RNA was tested using 1% agarose gel electrophoresis and a microvolume spectrophotometer (JP-Blupad1000, JinPeng Analytical Instruments Co., Ltd., Shanghai, China). First-strand cDNA synthesis was performed using the PrimeScript™ RT Reagent Kit with gDNA Eraser (Takara, RR047A, Takara Biomedical Technology Co., Ltd., Beijing, China) according to the manufacturer’s protocol. The reverse transcription of miRNA and synthesis of cDNA were performed using the miRNA 1st Strand cDNA Synthesis Kit (Cat: MR101-01, Vazyme Biomedical Technology Co., Ltd., Nanjing, China), following the manufacturer’s instructions. RT-qPCR was carried out using the 2× Taq Pro Universal SYBR qPCR Master Mix (Cat: Q712-03, Vazyme Biomedical Technology Co., Ltd., Nanjing, China). The PCR conditions were as follows: Denaturation at 95 °C for 30 s, denaturation at 94 °C for 5 s, annealing and extension at 60 °C for 30 s. All genes were analyzed in three biological replicates, with each biological replicate containing three technical replicates. The 2^−ΔΔCt^ method was used to calculate the mRNA expression levels of the genes.

### 4.8. Data Analysis

Data were processed using Excel 2019 and GraphPad Prism 8, and figures were generated using Origin 2021, on the Bioinformatics website (https://www.bioinformatics.com.cn). Two-way ANOVA was performed to evaluate the significance of differences in various parameters between the control group and different waterlogging durations. For Student’s *t*-test, *p* < 0.05, *p* < 0.01, *p* < 0.001, and *p* < 0.0001 were considered statistically significant (*), highly significant (**), very highly significant (***), and extremely significant (****), respectively.

## Figures and Tables

**Figure 1 plants-14-01340-f001:**
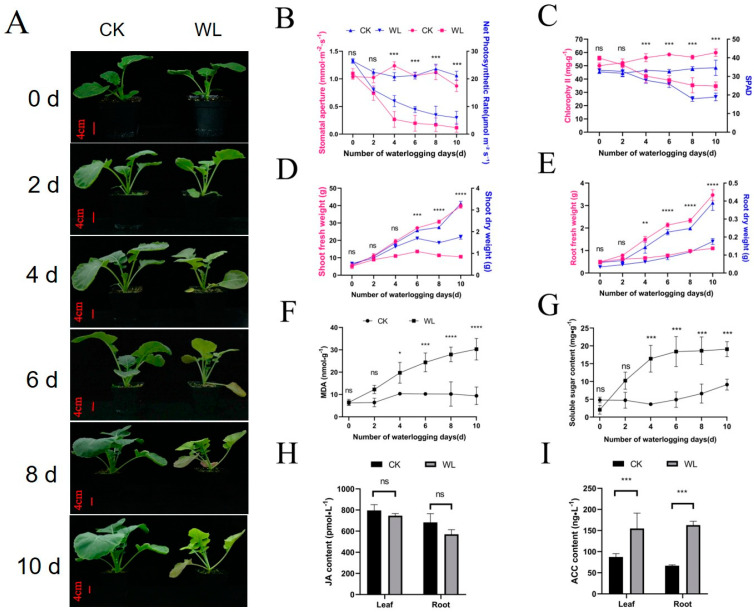
Morphological and physiological changes in rapeseed under waterlogging. (**A**) Morphological responses of ZS11 to waterlogging stress. Scale bar = 4 cm. (**B**) Waterlogging-induced reduction in leaf net photosynthetic rate and stomatal conductance. (**C**) Decline in leaf chlorophyll content and SPAD values under waterlogging. (**D**) Decreased aboveground biomass (fresh and dry weight) under waterlogging. (**E**) Reduced underground biomass (fresh and dry weight) due to waterlogging. (**F**) Accumulation of leaf MDA content under waterlogging stress. (**G**) Increased soluble sugar content in leaves under waterlogging. (**H**) Jasmonic acid accumulation in leaves and roots after 8 days of waterlogging. (**I**) ACC content (ethylene precursor) in leaves and roots after 8 days of waterlogging. Note: In the figure, “ns” indicates no significant difference, “*” represents significant differences from the CK at the same time point (*p* < 0.05), “**” indicates extremely significant differences (*p* < 0.01), “***” indicates *p* < 0.001, and “****” indicates *p* < 0.0001. CK: control; WL: waterlogging.

**Figure 2 plants-14-01340-f002:**
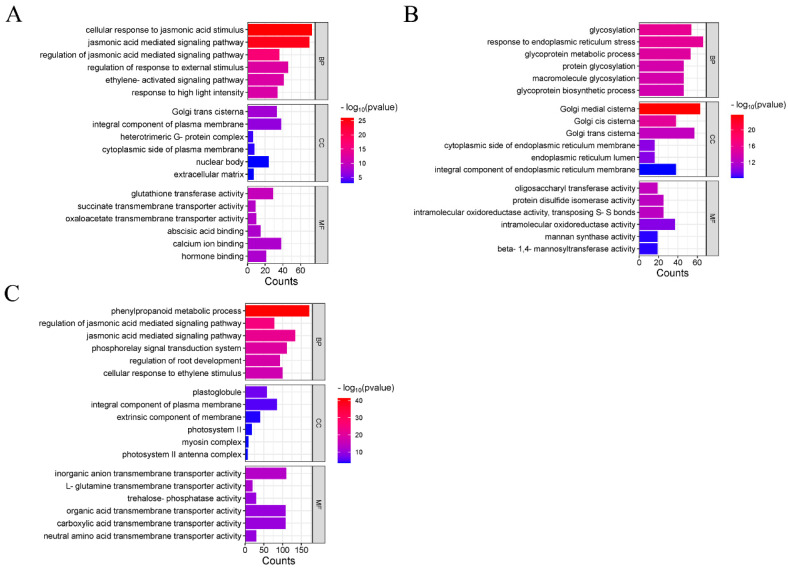
GO enrichment analysis of DEGs identified in roots and leaves. (**A**) GO enrichment analysis of common DEGs in both leaves and roots. (**B**) GO enrichment analysis of leave-specific DEGs. (**C**) GO enrichment analysis of root-specific DEGs.

**Figure 3 plants-14-01340-f003:**
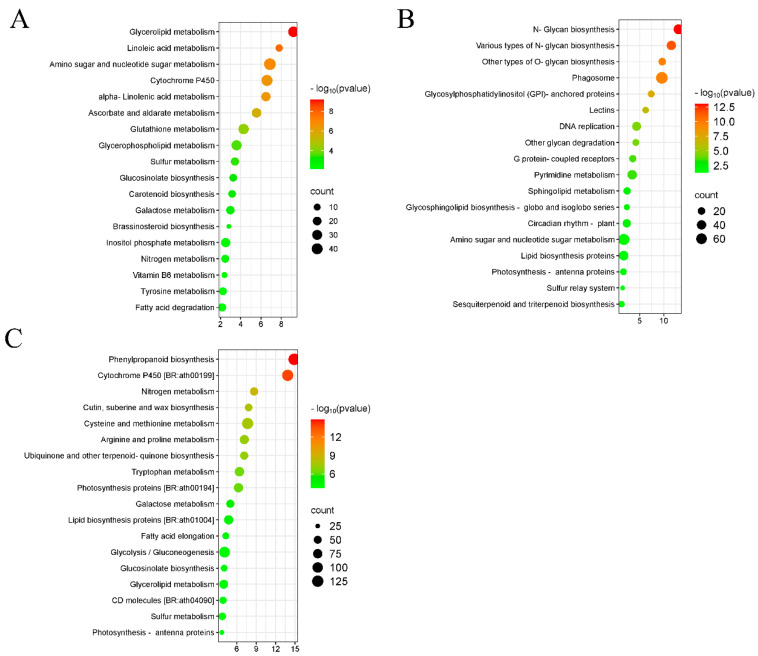
KEGG pathway classification of DEGs identified in roots and leaves. (**A**) KEGG pathway of common DEGs in both leaves and roots. (**B**) KEGG pathways of leave-specific DEGs. (**C**) KEGG pathways of root-specific DEGs.

**Figure 4 plants-14-01340-f004:**
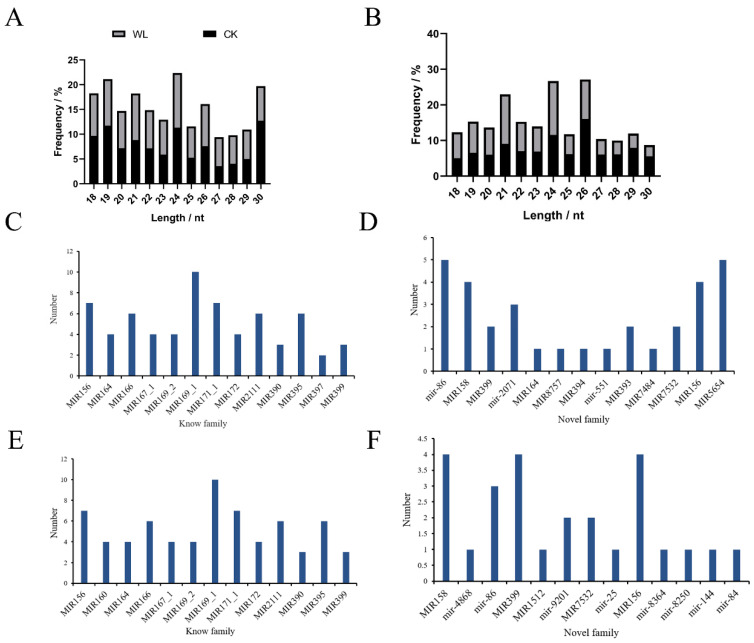
Base length distribution of miRNAs and miRNA families in leaves and roots. (**A**) miRNAs length distribution in roots. (**B**) miRNAs length distribution in leaves. (**C**) Length profiles of known miRNA families in leaves. (**D**) Length profiles of novel miRNA families in leaves. (**E**) Length profiles of known miRNA families in roots. (**F**) Length profiles of novel predicted miRNA families in roots.

**Figure 5 plants-14-01340-f005:**
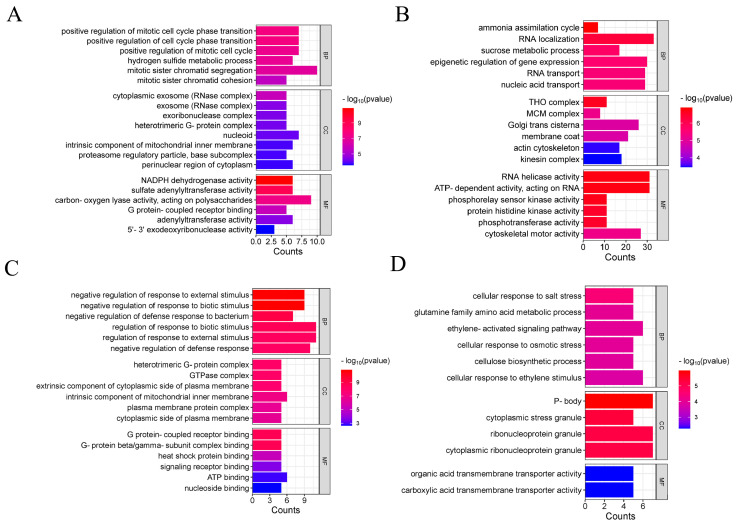
GO enrichment analysis of DEGs targeted by DEmiRNAs. (**A**) GO enrichment analysis of DEGs targeted by DEmiRNAs in leaves. (**B**) GO enrichment analysis of DEGs targeted by DEmiRNAs in roots. (**C**) GO enrichment analysis of DEGs targeted by novel-miR-23108 in leaves. (**D**) GO enrichment analysis of DEGs targeted by novel-miR-42624 in roots.

**Figure 6 plants-14-01340-f006:**
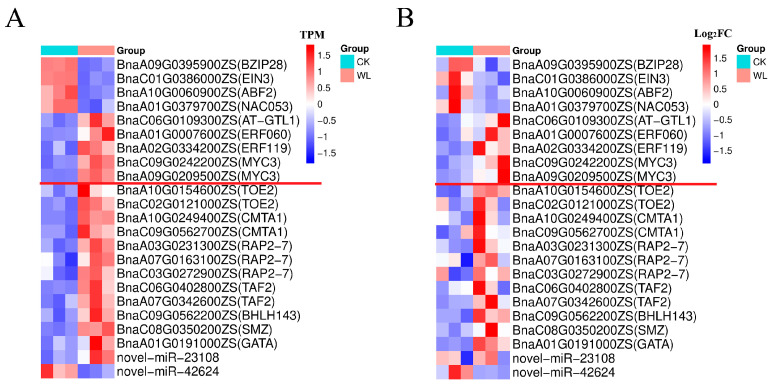
Expression analysis of the 21 TFs and the 2 novel miRNAs. (**A**) Transcriptome-based expression profiles of the 21 TFs and the 2 novel miRNAs. (**B**) RT-qPCR validation of TF expression levels, stem-loop RT-qPCR validation of miRNA expression levels. Note: In the figure, the nine TFs listed above the red line were identified in roots, while the twelve TFs below the red line were identified in leaves.

**Figure 7 plants-14-01340-f007:**
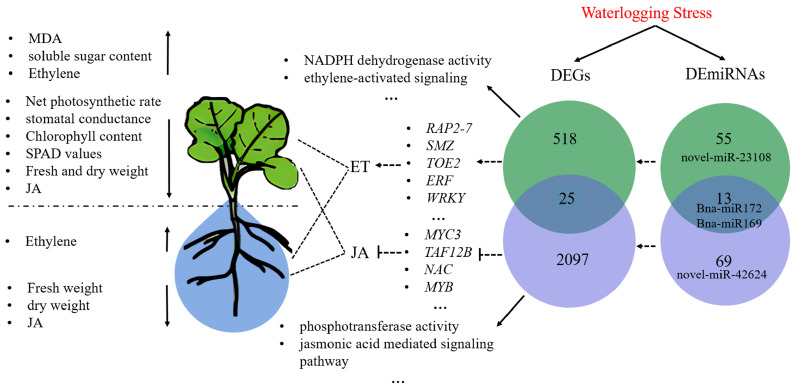
Schematic diagram of gene expression regulation in rapeseed under waterlogging stress.

**Table 1 plants-14-01340-t001:** The miRNA sequencing results of the *B*. *napus* library.

Read Type	Total	rRNA	snRNA	tRNA	Repeat	Other
Leaf-CKT-1	13,884,720	7,556,783 (54.43%)	48,196 (0.35%)	2,503,157 (18.03%)	2,709,768 (19.52%)	3,982,761 (28.68%)
Leaf-CKT-2	5,711,607	3,228,497 (56.53%)	14,078 (0.25%)	942,867 (16.51%)	1,155,668 (20.23%)	1,616,082 (28.29%)
Leaf-CKT-3	8,545,681	4,984,691 (58.33%)	39,373 (0.46%)	1,167,813 (13.67%)	1,785,472 (20.89%)	2,484,058 (29.07%)
Leaf-WLT-1	9,222,335	4,444,472 (48.19%)	60,181 (0.65%)	1,022,050 (11.08%)	2,030,677 (22.02%)	3,743,626 (40.59%)
Leaf-WLT-2	10,565,031	6,220,196 (58.88%)	60,938 (0.58%)	854,937 (8.09%)	2,587,181 (24.49%)	3,501,670 (33.14%)
Leaf-WLT-3	32,933,054	18,135,516 (55.07%)	310,346 (0.94%)	3,369,992 (10.23%)	8,104,513 (24.61%)	11,266,420 (34.21%)
Root-CKT-1	5,556,492	2,329,546 (41.92%)	16,652 (0.3%)	1,033,337 (18.60%)	1,392,040 (25.05%)	2,168,195 (39.02%)
Root-CKT-2	4,516,520	1,701,149 (37.67%)	15,000 (0.33%)	879,215 (19.47%)	1,064,131 (23.56%)	1,911,908 (42.33%)
Root-CKT-3	6,060,957	2,413,813 (39.83%)	22,376 (0.37%)	1,070,520 (17.66%)	1,416,834 (23.38%)	2,541,770 (41.94%)
Root-WLT-1	10,045,342	6,463,385 (64.34%)	25,259 (0.25%)	1,141,681 (11.37%)	3,856,248 (38.39%)	2,400,758 (23.90%)
Root-WLT-2	13,201,016	8,285,263 (62.76%)	35,380 (0.27%)	1,473,133 (11.16%)	5,008,072 (37.94%)	3,378,301 (25.59%)
Root-WLT-3	11,927,136	7,563,060 (63.41%)	27,194 (0.23%)	1,461,914 (12.26%)	4,629,966 (38.82%)	2,861,382 (23.99%)

## Data Availability

The raw sequence data generated in this study have been deposited in the Genome Sequence Archive [74] in National Genomics Data Center [75], China National Center for Bioinformation/Beijing Institute of Genomics, Chinese Academy of Sciences (GSA: CRA023479) that are publicly accessible at https://ngdc.cncb.ac.cn/gsa (accessed on 9 March 2025).

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
