# Peer review of "Transcriptome and Small-RNA Sequencing Reveals the Response Mechanism of Brassica napus to Waterlogging Stress"

_plants, 2025, doi:10.3390/plants14091340_

Round 1
Reviewer 1 Report
Comments and Suggestions for Authors
Waterlogging stress significantly impacts plant growth and development. The manuscript investigated the physiological and biochemical alterations in Brassica napus leaves and roots under waterlogging stress through transcriptome and small-RNA sequencing. The study preliminarily revealed the response mechanism of Brassica napus response to waterlogging stress regulated by miRNAs, providing a foundation for enhancing waterlogging tolerance in Brassica napus. The following revisions are necessary:
- The introduction must be improved as there is a lack of previous research on the effects of waterlogging stress on the growth and development of Brassica napus, as well as the response mechanisms of Brassica napus. Moreover, it lacks logic, the introduction should not only provide a brief introduction to ethylene, Jasmonic acid, root, leaf, and miRNAs mentioned in the manuscript.
- Figure annotations need improvement. In Figure 1, labels A-G should only indicate the measured parameters, omitting result statements. The annotations for "ns" in Figure 1 and "ns and ***" in Figure S1 are missing. Clarify whether Figure 1D represents above-ground parts or shoots.
- Why was the MDA content in roots not measured.
- The assessment of Jasmonic acid and ethylene content at only the eighth-day time point is insufficient.
- The discussion section needs expansion; the current discussion is overly concise.
- Figure 6 requires improvement; the current regulatory mechanism depiction is overly simplistic and lacks adequate summarization of the results.
- Indicate the dates used of the GO and KEGG database.
- Clarify whether all primers in Table S13 were designed by the authors or if some were sourced from the literature.
- The "4.8 Data Analysis" section needs revision to include details on the replication of parameters, the methods used for significance testing and multiple comparisons, the software utilized, and the significance level.
- In my personal opinion, consider moving fundamental data presented in Figures 2 and 4 to the supplementary materials, while incorporating key data from Figures S1, S3, S4, and S7 into the main text.
Author Response
|
Comments 1: The introduction must be improved as there is a lack of previous research on the effects of waterlogging stress on the growth and development of Brassica napus, as well as the response mechanisms of Brassica napus. Moreover, it lacks logic, the introduction should not only provide a brief introduction to ethylene, Jasmonic acid, root, leaf, and miRNAs mentioned in the manuscript. |
||||||||||||||
|
Response 1: Thank you for pointing this out. And we have incorporated recent advances in rapeseed waterlogging response research into the Introduction, and we have also restructured the sequence of the Introduction section to enhance its logical flow. This revision can be found on page 2, paragraph 2, line 50-61; page 2, paragraph 5, line 86-90. |
||||||||||||||
|
||||||||||||||
|
||||||||||||||
|
4. Response to Comments on the Quality of English Language |
||||||||||||||
|
We appreciate the reviewers' and editor’s comments regarding the English language quality. In the revised manuscript, we have carefully revised the language throughout the text to improve clarity and readability. The manuscript has been thoroughly checked and polished by a professional native-English-speaking editor to ensure the language meets the standards for publication. |

Reviewer 2 Report
Comments and Suggestions for Authors
A review of the manuscript entitled "Transcriptome and small-RNA sequencing reveals the response mechanism of Brassica napus to waterlogging stress".
The subjects addressed in the document pertain to the disciplines of plant physiology, molecular biology, and agriculture, particularly within the framework of climate change, which has led to an escalating prevalence of waterlogging stress. The authors concentrated on Brassica napus (rapeseed), which is of great economic importance. The employment of integrated transcriptomic and microRNA analyses facilitates the elucidation of multilevel mechanisms governing gene expression regulation in response to waterlogging stress.
The manuscript presents original results based on transcriptome analysis and small RNA sequencing comparing leaf and root responses. While analogous studies have been conducted in the past, the integration of microRNA and target gene expression data in rapeseed under waterlogging conditions remains a relatively under-explored domain. The manuscript's most significant contribution is its integration of a regulatory model that incorporates the roles of ethylene and jasmonic acid, which enhances the overall value of the work.
However, the manuscript would benefit from the following changes:
In the abstract, it would be beneficial to condense the descriptive section on the number of DEGs and DEmiRNAs and to emphasize the significance of these results more strongly.
Transcriptomic analyses are presented comprehensively. The significant number of differentially expressed genes (DEGs)—more than 12,000 in leaves and almost 20,000 in roots—raises concerns about the potential for overestimation of results (false positives). However, the methods used (DESeq2, log2FC > 1, p < 0.05) are standard.
The GO and KEGG enrichment analyses are robust. However, there is room for improvement in the presentation, as it is somewhat too synthetic in places. For instance, the metabolic pathways listed are not consistently interpreted in the context of plant biology. A more detailed assignment of the biological functions of individual differentially expressed genes would have been desirable.
The discussion is methodically structured, with citations to relevant literature. The authors appropriately accentuate the significance of hormones, such as ethylene and jasmonic acid, as well as selected miRNA families, namely miR172, miR169, and miR399. A notable strength of the study lies in the proposed regulatory model and its integration with transcriptomic data and physiological analysis.
Nevertheless, it would be beneficial to situate the findings within the framework of rapeseed breeding, particularly in the context of marker-assisted selection. A more extensive discussion is necessary to support the analysis of the differences between the root and the leaf, which is overly synthetic.
The entire manuscript requires linguistic revision by a native speaker, especially in the areas of syntax, grammatical tense, and preposition usage.
Author Response
|
3. Point-by-point response to Comments and Suggestions for Authors Comments 1: In the abstract, it would be beneficial to condense the descriptive section on the number of DEGs and DEmiRNAs and to emphasize the significance of these results more strongly. Response 1: Thank you for your suggestions. We have revised the content of the Abstract. This revision can be found on page 1, paragraph 1, lines 17–37.
Comments 2: Transcriptomic analyses are presented comprehensively. The significant number of differentially expressed genes (DEGs)—more than 12,000 in leaves and almost 20,000 in roots—raises concerns about the potential for overestimation of results (false positives). However, the methods used (DESeq2, log2FC > 1, p < 0.05) are standard. Response 2: Thank you for pointing out this. The methods and criteria we emplyed (DESeq2, log2FC > 1, p < 0.05) to identify DEGs are widely recognized as both rigorous and commonly used in transcriptomic research. Given that samples were collected on the 8th day of waterlogging stress—a stage when rapeseed growth was already significantly inhibited, and the WL and the CK have shown an obvious growth differences, this likely explains the substantial number of DEGs detected.
Comments 3: The GO and KEGG enrichment analyses are robust. However, there is room for improvement in the presentation, as it is somewhat too synthetic in places. For instance, the metabolic pathways listed are not consistently interpreted in the context of plant biology. A more detailed assignment of the biological functions of individual differentially expressed genes would have been desirable. Response 3: Thank you for your valuable comments. We have provided a more detailed description of several pathways identified in the GO and KEGG enrichment analyses. This revision can be found on page 5, paragraph 1, lines 155–160 and 163-166, page 5-6, lines 173–176.
Comments 4: The discussion is methodically structured, with citations to relevant literature. The authors appropriately accentuate the significance of hormones, such as ethylene and jasmonic acid, as well as selected miRNA families, namely miR172, miR169, and miR399. A notable strength of the study lies in the proposed regulatory model and its integration with transcriptomic data and physiological analysis. Nevertheless, it would be beneficial to situate the findings within the framework of rapeseed breeding, particularly in the context of marker-assisted selection. A more extensive discussion is necessary to support the analysis of the differences between the root and the leaf, which is overly synthetic. Response 4: Thank you for your valuable comments. We have thoroughly discussed the differences between roots and leaves, and also discussed the integration of our research findings with rapeseed breeding of waterlogging tolerance improvement. This revision can be found on page 11, paragraph 3, lines 333–339 and page 12-13, lines 389–394.
|
|
4. Response to Comments on the Quality of English Language |
|
We appreciate the reviewers' and editor’s comments regarding the English language quality. In the revised manuscript, we have carefully revised the language throughout the text to improve clarity and readability. The manuscript has been thoroughly checked and polished by a professional native-English-speaking editor to ensure the language meets the standards for publication. |

Reviewer 3 Report
Comments and Suggestions for Authors
The study provides a comprehensive analysis of the transcriptomic and small RNA responses of Brassica napus to waterlogging stress, identifying key miRNAs, target genes, and regulatory networks. The work is well-designed and contributes valuable insights into the molecular mechanisms underlying waterlogging tolerance. However, several areas could be strengthened to enhance the robustness and impact of the findings.
1.The study identifies novel miRNAs (e.g., novel-miR-23108, novel-miR-42624) and their predicted targets. Functional validation is critical to confirm their roles. Transient overexpression/silencing of key miRNAs (e.g., bna-miR172, bna-miR169) in rapeseed seedlings followed by phenotypic and physiological assays (e.g., root architecture, chlorophyll content, ethylene/JA levels) under waterlogging.
2.The study highlights ethylene and JA as key players but lacks mechanistic evidence for their interplay. Treat plants with ethylene (ACC) and JA (MeJA) inhibitors/analogs under waterlogging to dissect their roles in miRNA-mediated responses.
3.Roots show more DEGs than leaves, but the study lacks in situ validation of root-specific miRNAs/TFs. Histochemical staining (e.g., GUS reporters for miRNA promoters) to localize miRNA expression in roots.
4.The novel miRNAs (novel-miR-23108, novel-miR-42624) are predicted to target many DEGs but lack experimental confirmation. Stem-loop RT-qPCR to verify their expression patterns.
5.Compare findings with similar studies in other crops (e.g., cucumber, maize) to highlight conserved/divergent pathways.
6.Address why root-specific DEGs are more abundant than leaf-specific ones—is this due to tissue sensitivity or sampling timing?
7.Clarify the rationale for choosing 8-day waterlogging: Was this based on preliminary data showing peak stress responses?
8.Provide details on miRNA prediction criteria (e.g., fold-change thresholds, p-value adjustments).
Author Response
|
3. Point-by-point response to Comments and Suggestions for Authors |
|
Comments 1: The study identifies novel miRNAs (e.g., novel-miR-23108, novel-miR-42624) and their predicted targets. Functional validation is critical to confirm their roles. Transient overexpression/silencing of key miRNAs (e.g., bna-miR172, bna-miR169) in rapeseed seedlings followed by phenotypic and physiological assays (e.g., root architecture, chlorophyll content, ethylene/JA levels) under waterlogging. |
|
Response 1: Thank you for your suggestions. We agree with you that it is better to validate the function of miRNA, especially the novel miRNAs in this study. This study is one of the very few studies to date investigating rapeseed waterlogging response through miRNA analysis. Our results establish a foundation for understanding the pivotal role of miRNAs in rapeseed waterlogging response. And follow-up studies will comprehensively validate the functions of the identified DEmiRNAs, though the current research is primarily focused on the discovery and characterization of these DEmiRNAs. For DEmiRNAs with known functions identified in this study, like bna-miR172 and bna-miR169, we thoroughly discussed their functions in waterlogging response in the introduction section. |
|
|
|
Comments 2: The study highlights ethylene and JA as key players but lacks mechanistic evidence for their interplay. Treat plants with ethylene (ACC) and JA (MeJA) inhibitors/analogs under waterlogging to dissect their roles in miRNA-mediated responses. |
|
Response 2: Thank you for your valuable comments and suggestions. We agree that the hormone treatment experiments you suggested would provide more direct evidence of their functions in the waterlogging response of rapeseed. The roles of ethylene and jasmonic acid in waterlogging tolerance have been previously reported in other plants, such as soybean and cucumber. And we provided a through discussion of ET and JA on waterlogging response on page 12, paragraph 3, lines 369–376. Although time may be insufficient to include the corresponding experiments at this stage, we will incorporate relevant data in our future studies.
Comments 3: Roots show more DEGs than leaves, but the study lacks in situ validation of root-specific miRNAs/TFs. Histochemical staining (e.g., GUS reporters for miRNA promoters) to localize miRNA expression in roots. Response 3: We agree that in situ validation, such as histochemical staining using GUS reporters driven by root-specific miRNA promoters, would provide more direct evidence of miRNA spatial expression patterns. These experiments are part of our ongoing work and will be pursued in future studies to further validate and extend our findings. Regrettably, we were unable to incorporate these additional experimental results within the timeframe of this study.
Comments 4: The novel miRNAs (novel-miR-23108, novel-miR-42624) are predicted to target many DEGs but lack experimental confirmation. Stem-loop RT-qPCR to verify their expression patterns. Response 4: Thank you for your suggestions. We have analyzed the expression patterns of these two novel miRNAs, and the results are consistent with the small RNA sequencing data. Specifically, novel-miR-23108 was upregulated, and novel-miR-42624 was downregulated, as confirmed by our Stem-loop RT-qPCR analysis. This revision can be found on page 11, lines 306, figure 6.
Comments 5: Compare findings with similar studies in other crops (e.g., cucumber, maize) to highlight conserved/divergent pathways. Response 5: Thank you for your suggestions. In the discussion section, we incorporated studies on waterlogging responses across different crop species and identified several conserved response pathways through comparative analysis. This revision can be found on page 11, paragraph 3, lines 333–339.
Comments 6: Address why root-specific DEGs are more abundant than leaf-specific ones—is this due to tissue sensitivity or sampling timing? Response 6: Thanks for your suggestions. Waterlogging stress exerts a direct impact on rapeseed roots, while its effect on leaf growth is indirect, which may explain why a greater number of root-specific DEGs were identified. We have addressed this in the discussion section and proposed possible explanations. This revision can be found on page 11, paragraph 3, lines 327–333.
Comments 7: Clarify the rationale for choosing 8-day waterlogging: Was this based on preliminary data showing peak stress responses? Response 7: Based on biomass measurements and physiological-biochemical assays, rapeseed growth at the seedling stage was significantly inhibited when subjected to waterlogging for more than 4 days. After 8 days of waterlogging, the changes in some indicators began to stabilize, suggesting that the rapeseed plants had likely reached their peak response to waterlogging stress at this stage. And we have explained this in the discussion section (page 11, paragraph 1, lines 314–318).
Comments 8: Provide details on miRNA prediction criteria (e.g., fold-change thresholds, p-value adjustments). Response 8: Thank you for your valuable comments. We used the R package DESeq for statistical analysis and identified DEmiRNAs based on an adjusted p-value < 0.05 and |log₂(fold change, FC)| ≥ 1. We have revised the criteria for miRNA prediction. This revision can be found on page 14, paragraph 4, lines 467–469.
4. Response to Comments on the Quality of English Language |
|
We appreciate the reviewers' and editor’s comments regarding the English language quality. In the revised manuscript, we have carefully revised the language throughout the text to improve clarity and readability. The manuscript has been thoroughly checked and polished by a professional native-English-speaking editor to ensure the language meets the standards for publication. |
|
5. Additional clarifications |
|
Thank you again for your professional and specific comments and suggestions. We have diligently incorporated all feasible revisions into our manuscript. Regrettably, some experimental validations recommended in comments 1-3 would require substantial additional time to complete, and we currently cannot provide comprehensive explanations for these specific points. The present study primarily establishes the foundational framework for identifying waterlogging-responsive miRNAs and reconstructing regulatory networks in rapeseed. While these findings demonstrate novel insights, we fully acknowledge certain limitations that necessitate more extensive investigations in future studies. Our immediate research priorities will focus on systematic functional characterization of key miRNAs and DEGs; elucidating the synergistic mechanisms of ethylene and jasmonic acid signaling during waterlogging stress responses. We trust that our revisions have satisfactorily addressed the majority of your concerns, and we hope this additional clarification further strengthens the transparency and integrity of our study |

Round 2
Reviewer 1 Report
Comments and Suggestions for Authors
The current manuscript can be accepted by Plants.
Reviewer 3 Report
Comments and Suggestions for Authors
The author has completed all revisions. It is recommended to accept.